

**Measurement Report: Wintertime new particle formation in the rural area of North**
**China Plain: influencing factors and possible formation mechanism**

**Juan Hong[1,2†*], Min Tang[1,2†], Qiaoqiao Wang[1,2*], Nan Ma[1,2], Shaowen Zhu[1,2], Shaobin**
**Zhang[1,2], Xihao Pan[1,2], Linhong Xie[1,2], Guo Li[3], Uwe Kuhn[3], Chao Yan[4], Jiangchuan**
**Tao[1,2], Ye Kuang[1,2], Yao He[1,2], Wanyun Xu[5], Runlong Cai[6], Yaqing Zhou[1,2], Zhibin Wang[7],**
**Guangsheng Zhou[5], Bin Yuan[1], Yafang Cheng[3], Hang Su[3]**
[1]Institute for Environmental and Climate Research, Jinan University, Guangzhou, Guangdong
511443, China
[2]Guangdong-Hongkong-Macau Joint Laboratory of Collaborative Innovation for Environmental
Quality, Guangzhou, China
[3]Multiphase Chemistry Department, Max Planck Institute for Chemistry, Mainz 55128, Germany
[4]School of Atmospheric Sciences, Joint International Research Laboratory of Atmospheric and
Earth System Sciences, Nanjing University, Nanjing, China
[5]Hebei Gucheng, Agrometeorology, National Observation and Research Station, Chinese Academy
of Meteorological Sciences, Beijing, 100081, China
[6]Institute for Atmospheric and Earth System Research/Physics, Faculty of Science, University of
Helsinki, Helsinki, FI00014, Finland
[7]College of Environmental and Resource Sciences, Zhejiang University, Zhejiang Provincial Key
Laboratory of Organic Pollution Process and Control, Hangzhou 310058, China
[†]These authors contributed equally to this work.
[*]Correspondence: Qiaoqiao Wang (qwang@jnu.edu.cn) and Juan Hong
(juanhong0108@jnu.edu.cn)
**Abstract:**
The high concentration of fine particles as well as gaseous pollutants makes
polluted areas, such as the urban setting of North China Plain (NCP) of China, a
different environment for NPF compared to many clean regions. Such conditions also
hold for other polluted environments in this region, for instance, the rural area of NCP,
yet the underlying mechanisms for NPF remain less understood owing to the limited
observations of particles in the sub-3nm range. Comprehensive measurements,
particularly covering the particle number size distribution down to 1.34 nm, were
conducted at a rural background site of Gucheng (GC) in the North China Plain (NCP)
from 12 November to 24 December in 2018. Five NPF events during the 39 effective
days of measurements for the campaign were identified, with the mean particle





nucleation rate ($J_{1.34}$) and growth rate (GR$_{1.34-2.4}$) were 29.1 cm$^{-3}$·s$^{-1}$ and 0.54 nm·h$^{-1}$,
respectively. During these five days, NPF concurrently occurred in an urban site in
Beijing, indicating that NPF events during these days in this region might be a regional
phenomena. This implies that H2SO4-amine nucleation, concluded for urban Beijing
there, could also be the dominating mechanism for NPF at our rural site. The
condensation sink or coagulation sink for the survival of newly-formed and small
clusters are the dominating factor controlling the occurrence of NPF under current
atmosphere, whereas the contribution from the available H2SO4 cannot be neglected,
either. This feature is slightly different from that of urban Beijing, where CS mainly
determines whether NPF takes place or not.

**Keywords:** new particle formation, particle number size distribution, condensation
sink, nucleation mechanism.





## 1. Introduction


Atmospheric new particle formation (NPF) is a major source of the global particles
in terms of number concentration and size distribution (Kulmala et al., 2004) and is
considered to contribute up to half of the global cloud condensation nuclei (CCN)
budget in the lower troposphere (Spracklen et al., 2006; Dunne et al., 2016). In general,
NPF consists of two consecutive processes: a) the formation or nucleation of molecular
clusters by low-volatile gaseous substances, and b) their subsequent growth to
detectable sizes or even larger, at which these particles may act as CCNs or contribute
to the particle mass concentration (Kulmala et al., 2000; Zhang et al., 2012).
Numerous laboratory measurements and field studies have shown that sulfuric
acid ($H_2SO_4$) are the key precursors to form molecular clusters for nucleation
(Nieminen et al., 2010; Sipilä et al., 2010; Kirkby et al., 2011; Riccobono et al., 2014;
Stolzenburg et al., 2020). However, these $H_2SO_4$ clusters relevant to atmospheric
nucleation are typically quite small, i.e., with diameters below 1.5 nm, at which the
detection efficiency of traditional instruments specific for NPF was usually
unsatisfactory (Kulmala et al., 2013). This had led to large uncertainties in the
measured formation rate of newly-formed particles and thus required precise
measurements of these clusters or particles down to sub-3 nm. Upon recently,
progress such as the use of a particle size magnifier (PSM) (Vanhanen et al., 2011; Xiao
et al., 2015), a neutral cluster and air ion spectrometer (NAIS) (Pushpawela et al., 2019;
Sulo et al., 2020) and a chemical ionization atmospheric pressure interface time of-
flight mass spectrometer (CI-APi-TOF) (Kürten et al., 2016; Sulo et al., 2020) make it
possible to directly measure the number concentration as well as the chemical
composition of clusters in the 1-3 nm size range. Benefit from these novel techniques,
observations have found that the growth of $H_2SO_4$ clusters would be significantly
promoted after stabilized by other precursors like amines, ammonia or iodine species
(Berndt et al., 2010; Kirkby et al., 2011; Almeida et al., 2013; Riccobono et al., 2014;
Kürten et al., 2016; Sipilä et al., 2010). Furthermore, oxidation products from volatile



organic compounds, for instance, highly oxidized organic compounds, were suggested
to be important contributors participating in atmospheric nucleation (Ehn et al., 2014;
Bianchi et al., 2016; Kirkby et al., 2016; Tröstl et al., 2016).
The North China Plain (NCP) of China, has been suffering heavily from the highly
complex air pollution since decades (Ma et al., 2016; Shen et al., 2018; Zhang et al.,
2020), owing to the high emissions or formations of different pollutants such as SO2,
NH3, VOCs as well as fine particles from various sources (Guo et al., 2014; Zhang et al.,
2015). Due to the high concentration of pre-existing particles, previous studies
considered that in the NCP, less NPF would occur as the newly-formed particles would
be scavenged much faster before growing. By contrast, atmospheric NPF was still
frequently observed in this region (Chu et al., 2019; Deng et al., 2020; Cai et al., 2021),
being more often than theoretically predicted (Kulmala et al., 2014), indicating that
the underlying mechanisms for NPF in this area might be different, that those
mechanism previously found for other environments might not be completely
applicable. The higher concentration of these gaseous precursors makes this region an
unique condition for NPF compared to relatively clean environments (Kulmama et al.,
2016; Yu et al., 2017; Wang et al., 2017), further supporting the hypothesis of different
formation mechanisms and thereby distinct features of NPF events in this region.
These doubts concerning NPF in the NCP, however, still remain to be elucidated due to
limitations of comprehensive measurements, particularly for rural areas of the NCP,
where observations regarding NPF was even more rare.
In addition, with respect to those existing studies concerning NPF in the NCP, they
mainly focused on the measurements of particles beyond 3 nm. Without applicable
instruments, observations of new particles down to sub-3nm was still quite limited
(Fang et al., 2020; Zhou et al., 2020), causing large uncertainties in the measured
characteristics of NPF for current region. To fill the gap of measurements of particles
or clusters in the size range of 1-3 nm and further advance our understanding of NPF
in this region, particularly in the rural area of NCP, we conducted a comprehensive
measurement campaign at a rural background site in the NCP during November 12 to
December 24, 2018. By obtaining the particle number size distribution over a wide



diameter range (1.34 nm - 10 μm), we aimed to investigate the characteristics of NPF
events at the rural site in NCP during wintertime, find out which factors govern the
occurring of NPF compared to other regions of NCP such as the urban areas and
explore the potential mechanisms for NPF in this area.



## 2. Experiment

### 2.1. Field measurements site

The measurements were conducted at Gucheng (GC) site (39°09'01.1"N 115°44'02.6"E), situated at an Ecological and Agricultural Meteorology Station (39°09' N, 115°44' E) of the Chinese Academy of Meteorological Sciences from 12 November to 24 December in 2018. The station is located in Dingxing county, Baoding city, Hebei Province, China, as seen in Fig.1 and surrounded by agricultural fields and sporadic villages. Being far from the urban and industrial emission areas, this site can be treated as a representative regional site in the northern part of NCP. More details about this site can be found in Lin et al. (2009) and Shen et al. (2018).

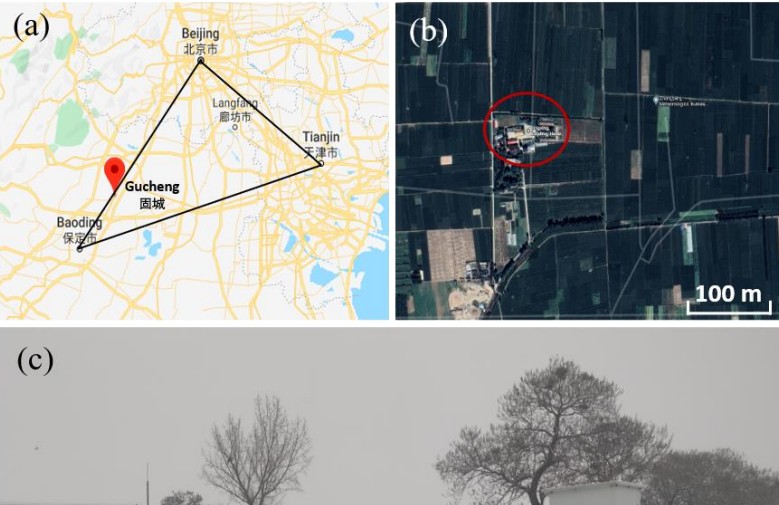

Figure 1. The upper panel shows the geographical location of the site (red dot and circled, © Google Maps), where our field measurements were carried out. The lower panel shows the measurement containers, where the sampling instruments were set up.


**2.2. Measurements**
**2.2.1.  Particle Number Size Distribution (PNSD) measurement**
The aerosol sampling inlet was located on the rooftop of a measurement
container, where room temperature was maintained at 22℃ (Fig1. c). The aerosol was
sampled via a low-flow PM10 cyclone inlet, passed through a Nafion dryer, and
directed to different instruments through stainless steel or conductive black tubings
using an isokinetic flow splitter. The particle number size distribution of aerosol
particles with diameters from 10 nm to 10000 nm was measured by using a scanning
mobility particle sizer (SMPS, model TSI 3938) and an Aerodynamic Particle Size
Spectrometer (APS, model TSI 3321) at a time resolution of around 5 minutes. The
SMPS consisted of an electrostatic classifier (model TSI 3080), a differential mobility
analyzer (DMA, model TSI 3081) and a condensation particle counter (CPC, model TSI

139 3772).

**2.2.2.  Sub-3nm Particle Number Concentration measurement**
Sub-3nm particles were measured with an Airmodus nano Condensation Nucleus
Counter system (nCNC, model A11), consisting of a Particle Size Magnifier (PSM, model
A10) and a butanol condensation particle counter (CPC, model A20) (Kangasluoma et
al., 2016). The Airmodus PSM uses diethylene glycol as the working fluid to activate
and grow nano-sized particles. Specifically, the PSM was operated under the scanning
mode that the diethylene glycol flow was varied between 0.1 to 1.3 L·min$^{-1}$. Thus, the
number size distribution of five different size bins, i.e., 1.34-1.39, 1.39-1.60, 1.60-1.94,
1.94-2.40, and 2.40-3.70 nm was obtained. Owing to the data quality, only the former
four size bins data were used in this study. During this campaign, the duration of each
scan was completed within around 240 s.
**2.2.3.  Pollutant gases, PM2.5 and meteorological parameters measurement**
Concentration of trace gases, including $SO_2$, $O_3$, CO and NOx, was measured



continuously during this campaign using different Themo Fisher Analyzers (model 43i-
TLE, 49i, 48i, and 42i), respectively, at a time resolution of 1 minute. The non-refractory
submicron aerosol chemical composition was measured by an Aerosol Chemical
Speciation Monitor (ACSM, Aerodyne, USA) (Sun et al., 2012) and the black carbon
mass concentration was measured by a 7-wavelength aethalometer (model AE-33,
Magee Scientific Inc., USA) (Petzold et al., 2013) using a PM2.5 inlet.
In addition, ambient meteorological conditions, such as wind speed, wind
direction, temperature, relative humidity and solar radiation, were also regularly
measured in another building, which is located about 20 meters to the southwest of
the container, at the same observational site.
**2.3. Data processing**
**2.3.1.   Formation Rate ($J_{Dp}$) and Growth Rate ($GR$)**
$J_{Dp}$ defines the formation rate of atmospheric particles at a certain diameter ($D_P$)
and can be calculated according to Kulmala et al. (2012) as:
$$J_{D_p} = \frac{dN_{\Delta D_p}}{dt} + CoagS_{\Delta D_p} \times N_{\Delta D_p} + \frac{1}{\Delta D_p} GR_{\Delta D_p} \times N_{\Delta D_p}$$
where $N$ is the particle number concentration between the diameter $dp_2$ and $dp_1$
(denotes as $\Delta D_P$), $CoagS$ is the coagulation sink of particles, $GR$ is the particle
growth rate out of the selected size bin.
In our study, we used two independent methods to calculate GR. One is the
maximum concentration method (Kulmala et al., 2012), being mainly for PSM data.
The other is based on the variation in geometric mean diameters of particle number
size distribution, which is derived by fitting the PNSD into 2 or 3 log-normal modes
using an automatic algorithm (DO-FIT model) (Hussein et al., 2005), mainly for SMPS
data.
$$GR = \frac{ddp}{dt} = \frac{\Delta dp}{\Delta t} = \frac{dp_2 - dp_1}{t_2 - t_1}$$
where $dp_1$ and $dp_2$ were particle diameters at time $t_1$ and $t_2$, respectively.





**2.3.2. Condensation Sink ($CS$) and Coagulation Sink ($CoagS$)**
$CS$ describes how fast the low-volatility molecules condense onto pre-existing
aerosols and can be expressed as (Kulmala et al., 2012):
$$CS = 2\pi D \int_{0}^{dpmax} \beta_{m,dp} dp N_{dp} \, ddp = 2\pi D \sum_{dp} \beta_{m,dp} dp N_{dp}$$

where $D$ is the diffusion coefficient of the condensing vapor, which is usually referred
to sulfuric acid and $\beta_{m,dp}$ is the mass flux transition correction factor.
$CoagS$ represents how fast the fresh formed particles are lost to pre-existing
particles through coagulation and can be calculated as :
$$CoagS_{dp} = \int K(dp, dp')n(dp) \, ddp' \cong \sum_{dp'=dp}^{dp'=max} K(dp, dp')N_{dp'}$$

where, $K\left(dp, dp'\right)$ is the collision efficiency between particles at the diameter
from $dp$ to $dp'$.
**2.3.3. Sulfuric Acid proxy ($SA$ proxy)**
SA was considered as one of the key precursors responsible for particle nucleation
in the atmosphere. However, no direct measurement for the concentration of SA was
available in current study. We therefore used a proxy variable to substitute the
concentration of SA, as SA is mainly produced by the oxidation of SO2 by OH radicals,
which can be approximated by the UV-B intensity (Petäjä et al., 2009). Thus, the proxy
concentration of SA can be calculated by (Zhu et al., 2017):
$$SA \, proxy = k \cdot \frac{[SO_2] \cdot SR}{CS}$$

where, $k$ is a scaling constant and was assumed to be $2.3 \times 10^{-9}$ m²/(W·s).
**2.3.4. Classification of NPF event**
Days of NPF events was classified according to the method proposed by Dal Maso
et al. (2005) and Kulmala et al. (2012), in which (a) a new particle mode appears from
the particle number size distribution within the nucleation mode size range, (b) the



particles in new mode prevail and have a continuous growth over a time span of hours.

In addition to the traditional classification for NPF, a burst in sub-3nm particles or

clusters and subsequent growth to larger size for a few hours that was visually available
from PSM data was also considered as an NPF event.





## 3. Results and discussions

### 3.1. General characteristics of NPF at GC site

Figure 2 shows the time series of meteorological parameters (a: wind speed and direction, b: temperature and relative humidity) and aerosol properties (c: total surface and volume concentration, d and e: PNSD in the size range of 10 to 800 nm and particle number concentration in the range of 1.34 to 2.40 nm) during this field campaign. During our study, wind speed was typically quite low with an average of 1.18 m·s$^{-1}$, indicating stagnant meteorological conditions for the limited dilution of air pollutants at current site. The temperature and relative humidity (RH) show opposite diurnal variation over the observational period, with the highest temperature and lowest RH during daytime and vice versa during nighttime.

According to the PNSD and PSM data, five days, with three of which having significant burst of sub-3 nm clusters as shown in Fig.2e, were classified as NPF event out of the total experimental period. This corresponds to an NPF frequency of 12.8%, which was lower compared to those at an urban site (ie., Beijing) in the same region during the same season (Shen et al. (2018) (25.8%); Deng et al. (2020) (51.4%)). Similar findings were also observed in Yue et al. (2009) and Wang et al. (2013), that NPF frequencies were higher at the Beijing urban site than at the corresponding regional background or rural site. They attributed this to the higher pollution level and correspondingly higher precursor content in the urban cities, leading to stronger NPF events there.



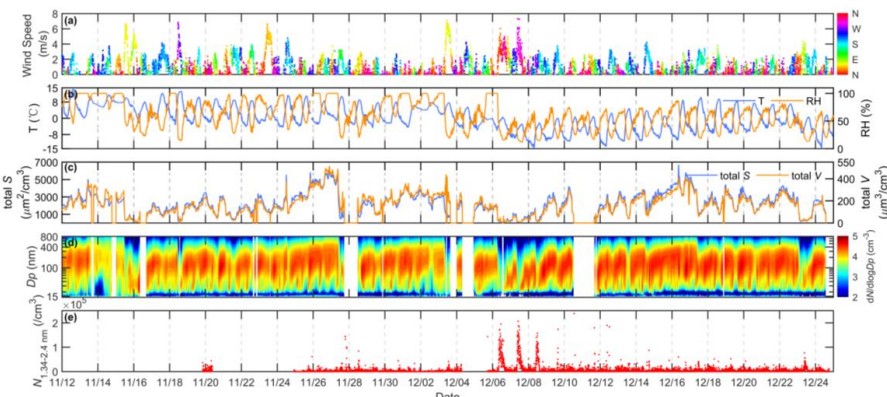

**Figure 2.** Time series of (a) wind speed and wind direction, (b) temperature (*T*) and relative
humidity (RH), (c) total particle surface and volume concentration calculated by using PNSD data,
(d) measured PNSD in the size range of 10 - 800 nm, (e) particle number concentration in the range
of 1.34 to 2.40 nm during the entire measurement period (2018.11.12-2018.12.24). White portion
indicates no data was available due to instrument maintenance or power failure.

Figure 3 shows a typical NPF event on December 7 as an example. Northwest wind
prevailed with elevated wind speed starting from around 8:00 o'clock, which was
conducive to the diffusion of local pollutants, leading to a dramatic decrease in CS
concurrently. At the same time, an obvious rise in H2SO4 concentration was observed,
coinciding with a strong burst in the concentration of sub-3 nm clusters. Then, new
particles with diameter larger than 10 nm, as shown in Fig. 3b, gradually formed by
growth, exhibited as a visible banana shape in PNSD.





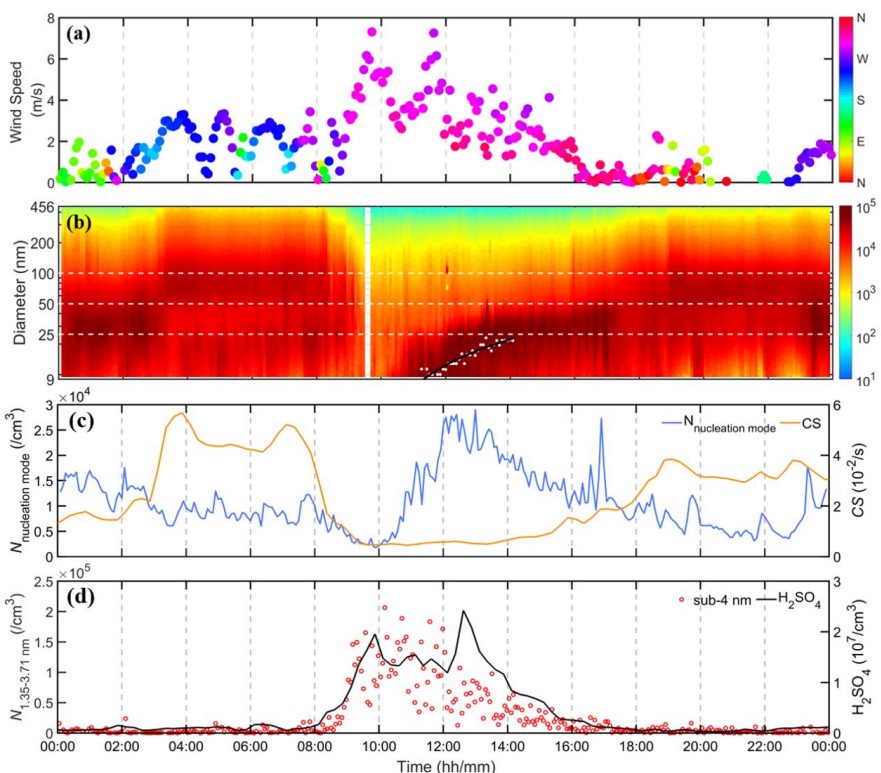


**Figure 3.** A case of NPF event on December 7 during this field campaign. Time series of (a) wind speed and wind directions, (b) the PNSD in the size range of 9-400 nm (The white dotted line represents the size with diameter at 25, 50, and 100 nm; black line represents the polynomial fit of the measured PNSD, (c) the particle number concentration of nucleation mode (9-25nm) and *CS*, (d) the number concentration of sub-3nm clusters and predicted concentration of sulfuric acid.

For all the identified NPF events, the formation rate of 1.34 nm ($J_{1.34}$) particles ranged from 8.01 cm$^{-3}$·s$^{-1}$ to about 40.8 cm$^{-3}$·s$^{-1}$ with an average value of 29.1 cm$^{-3}$·s$^{-1}$ at our GC site during the measurement period. It has to be noted that most atmospheric formation rate reported in China was based on the measured formation rate at relatively larger size, i.e., 3-10 nm. However, according to Chu et al. (2019), large errors may associate with the deviations of $J$ when using data at larger sizes as GR at sub-3 nm is needed but were typically unclear. Therefore, we focused more on the formation rate of particles at sizes below 3 nm in the following discussion. In principle, particle formation rate is inversely proportional to the CS, as the nucleation precursors or

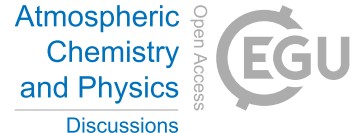
clusters would be scavenged more rapidly under higher CS conditions, leading to a
slower nanoparticle formation with a lower $J$. However, as shown in Table 1, in spite of
the higher CS, the particle formation rate at our site appears to be higher than those
in clean environments. This kind of intensive NPF becomes more noticeable for those
Chinese megacities, such as Shanghai, Beijing and Nanjing, having an even higher $J$ and
CS compared to that at our GC site. The most plausible explanation could be the more
abundance of nucleating precursors for NPF in those polluted atmosphere, which is
clearly proved by the SA concentration, either measured or calculated. To be specific,
the mean SA concentration during NPF at our GC site was around $7 \cdot 10^6$ cm$^{-3}$, a factor
of above 10-20 higher than that at Hyytiälä in Finland. The SA concentration during
NPF at Shanghai and Nanjing was even higher, being around $4 \cdot 10^7$ cm$^{-3}$.










**Table 1.** Summaries of the parameters (average value) relevant for NPF event during wintertime in China
and other countries.

| Station | Period | Frequency | $J$ (cm$^{-3}$·s$^{-1}$) | GR (nm·h$^{-1}$) | CS (10$^{-2}$·s$^{-1}$) | SA (10$^6$·cm$^{-3}$) | Reference |
|---|---|---|---|---|---|---|---|
| GC [R] | 2018.11.18 | - | 3.15 ($J_{10}$) | 4.27 | 4.7 | 4.1 | This study |
| GC [R] | 2018.12.06 | - | 39.4 ($J_{1.34}$) | 1.84 | 0.74 | 10.4 | This study |
| GC [R] | 2018.12.07 | - | 40.8 ($J_{1.34}$) | 4.05 | 0.8 | 11.3 | This study |
| GC [R] | 2018.12.08 | - | 28.2 ($J_{1.34}$) | 8.11 | 2.7 | 4.59 | This study |
| GC [R] | 2018.12.23 | - | 8.01 ($J_{1.34}$) | 1.23 | 1.6 | 5.35 | This study |
| GC [R] (mean) | 2018.11.12-12.24 | 12.8% | 29.1 ($J_{1.34}$) | 3.9 | 2.1 | 7.15 | This study |
| Thissio [UB] | 2015.8-2016.8, 2017.2-2018.2[a] | 10.3% | 1.55 ($J_{10}$) | 3.48 | 0.79 | 6.33 | (Kalkavouras et al., 2020) |
| New Delhi [U] | 2002.10.26-2002.11.9 | 53.3% | 7.3 ($J_3$) | 14.9 | 5.75 | - | (Mönkkönen et al., 2005) |
| Panyu [U] | Winter of 2011 | 21.3% | 0.89 ($J_{10}$) | 5.1 | 5.5 | - | (Tan et al., 2016) |
| Shanghai [U] | 2013.11.25-2014.1.25 | 21% | 188 ($J_{1.34}$) | 11.4 | 6.0 | 37 | (Xiao et al., 2015) |
| Nanjing [U] | 2011.11.18-2012.3.31 | 20% | 33.2 ($J_2$) | 8.5 | 2.4 | 45.3 | (Herrmann et al., 2014) |
| Hongkong [U] | 2010.10.25-2010.11.29 | 34.3% | 2.94 ($J_{5.5}$) | 3.86 | 0.8-6.2 | 9.17 | (H. Guo et al., 2012) |
| Beijing [U] | 2018.1.23-2018.3.31 | 51.5% | 38 ($J_{1.5}$) | 5.5 | 3.7 | 4.13 | (Chu et al., 2021) |
| Ziyang [R] | 2012.12.5-2013.1.5 | 23% | 5.2 ($J_3$) | 3.6 | 7.4 | 6.7 | (Chen et al., 2014) |
| Melpitz [R] | Winter of 2003-2006 | 3% | 0.7 ($J_3$) | 5.6 | 1.2 | 0.123 | (Hamed et al., 2010) |
| Melpitz [R] | Winter of 1996-1997 | 10% | 4.9 ($J_3$) | 4.1 | 0.9 | 0.259 | (Hamed et al., 2010) |
| Pingyuan [R] | 2017.11.3-2018.1.20 | 39.2% | 164.2 ($J_{1.37}$) | 3.9 | 1.9 | 2.45 | (Fang et al., 2020) |
| Xinken [R] | 2004.10.3-2004.11.5 | 25.9% | 0.5-5.4 ($J_3$) | 2.2.-19.8 | - | - | (Liu et al., 2008) |
| Solapur [R] | 2018.10-2019.2 | 28.9% | 0.22-10.07 ($J_{15}$) | 1.2-13.8 | 0.6-3 | - | (Varghese et al., 2020) |
| Cyprus [RB] | 2018.1-2018.2 | 69% | 16.4 ($J_{1.5}$) | 9.97 | 1.2 | - | (Baalbaki et al., 2020) |
| SEAS [O] | Winter of 2018 | 5% | 2.95 ($J_{10}$) | 14.35 | 4.5 | - | (Kompalli et al., 2020) |
| SMEAR II [B] | Winter of 1996-2003 | 24.2% | 0.2-1.1 ($J_3$) | 0.29-3.7 | 0.05-0.35 | 0.53 | (Dal Maso et al., 2005) |



SEAS: the southeastern Arabian Sea
R: rural site        UB: urban background site        RB: rural background site        U: urban site.        B: background site        O: ocean
site
a: only in wintertime        -: no number



Although the formation rate of 1.34 nm particles is relatively high, the newly-formed
particles at our GC site usually cannot grow into very large particles within a short time,
indicative by their low GR. The average value of $GR_{1.34-2.4}$ and $GR_{9-15}$ at our site was 0.54
nm/h and 3.9 nm/h, respectively, being generally lower than many clean environments
($GR_{1-3}$ of 0.9 nm/h for Hyytiälä, of 5.1 nm/h for Jungfraujoch), but similar to those at
urban Beijing and rural Pingyuan. This could be attributed by the high CS or CoagS at
those sites that small particles are vulnerable to the coagulation scavenging. However,
despite the high CoagS, the observed GR at Shanghai and Nanjing was still
exceptionally high. This discrepancy suggests that besides the high concentration of
precursors, mainly $H_2SO_4$, in polluted environments including both rural and urban
sites, other precursors with different efficiency for nanoparticle growth, and other
involving mechanisms, for instance, multiphase reactions, may all contribute to the
nanoparticle growth, yet to be elucidated.
**3.2. Potential mechanisms for NPF events in the rural NCP**
To further understand the dominating nucleation mechanism in the rural atmosphere
of NCP in China, we plotted the measured formation rate of 1.34 nm particles ($J_{1.34}$)
against the simulated $H_2SO_4$ concentration and compared the results to previous
studies conducted in different environments, as shown in Fig. 4. The formation rate of
particles under similar H2SO4 level for our study approximated most to the formation
rates measured by these CLOUD (The Cosmics Leaving OUtdoor Droplets chamber)
experiments using H2SO4 and DMA as the nucleating vapors. This suggested that
H2SO4 clustering drove the initial steps of NPF at our GC site, and the molecules
stabilizing H2SO4 clustering were most likely DMA.

For NPF in China, the $J_{1.34}$-$H_2SO_4$ relationship in our results were also close to that in
Beijing (Cai et al., 2021), an urban site in the NCP, but with a lower formation rate
under similar H2SO4 level. Cai et al. (2021) and Yan et al. (2021) concluded that H2SO4-
DMA was the dominating nucleation mechanism for urban Beijing with an additional



311 support from the measured C2-amine concentration. It has to be noted that their study

312 was conducted during a much longer time and completely covered the measurement

313 period of our study. More importantly, during the five days of events in our study, NPF

314 concurrently occurred at their measurement site (Liu et al., 2020). This suggests that

315 NPF events during these days in this region might be a regional phenomena, sharing

316 the same or similar nucleation mechanism. Therefore, we conclude that clustering of

317 H2SO4 with DMA may also dominate the nucleation process at our site during winter.

319 On the other hand, we noticed that our results deviates significantly from the

320 measured formation rate at Pingyuan (Fang et al., 2020), another rural site in the NCP.

321 They concluded that neither H2SO4-NH3 nor H2SO4-DMA mechanisms could fully

322 explain their observed particle formation rate but suggested that gaseous dicarboxylic

323 acids were the dominating species for the initial step of H2SO4 clustering under diacid-

324 rich environment. Being likewise the rural environment of NCP, we cannot completely

325 rule out the contribution of dicarboxylic acids to the H2SO4 stabilizing. However, by

326 taking into account the contribution from dicarboxylic acid (measured by a iodine-

327 based chemical ionization-atmospheric pressure interface-time-of-flight (I-APi-TOF,

328 Aerodyne Research Inc., USA)), the obtained $J_{1.34}$-$H_2SO_4$×diacids relationship was not

329 improved (see Fig. S1), being obviously different from the case of Pingyuan. Hence, the

330 involvements of diacids during the initial steps of nucleation under current rural

331 atmosphere cannot be confirmed, requiring future data, for instance, the signal of

332 H2SO4-diacids to be elucidated. This statement does not necessarily mean that our

333 previous inference was incorrect, but on the other side, provides some hints that

334 though NPF events in the NCP is regional, there might be no uniform theory but

335 multiple mechanisms coexisting to explain its feature with the dominating one varies

336 upon different emission patterns or meteorological conditions.




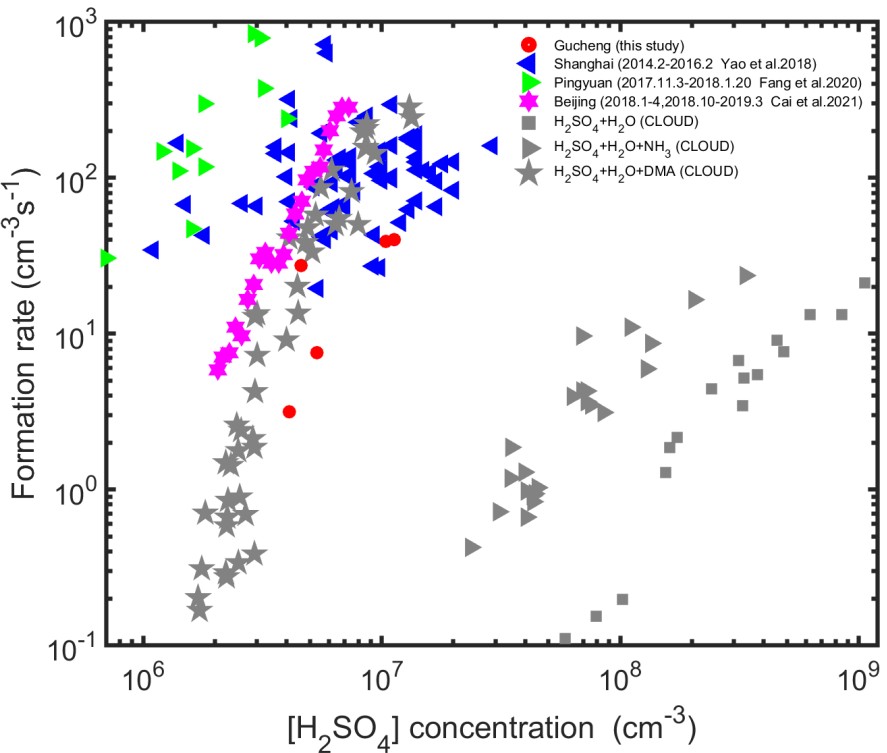


**Figure 4.** The particle formation rate ($J_{1.34}$) as a function of $H_2SO_4$ concentration for our study as well as for urban Shanghai , Beijing, rural Pingyuan and CLOUD measurements. Gray square, triangle, pentagram, and diamond represents the CLOUD data for $H_2SO_4+H_2O$, $H_2SO_4+H_2O+NH_3$, $H_2SO_4+H_2O+DMA$ (Kirkby et al. (2011) and Riccobono et al. (2014)), where DMA represents dimethylamine.

**3.3 Governing factors for the occurrence of NPF in rural NCP**

The high concentration of SO2, NH3, NOX, VOCs as well as fine particles makes the NCP of China an unique condition for NPF compared to many other environments. In principle, the competition between how fast the newly-formed clusters grow and how fast they are scavenged determines whether NPF will occur or not in the atmosphere. However, in the NCP, the concentration of SA was typically quite high, probably reaching its maximum rate to form clusters. Thus, CS or CoagS becomes the dominant factor controlling the occurrence of NPF. This was partly confirmed by existing



observations, for instance, Cai et al. (2021) found that H2SO4 was high enough in
urban Beijing, but not necessarily led to the occurrence of NPF there. They pointed out
that as long as CS or CoagS was below a certain threshold (Cai et al., 2017), NPF is very
likely take place.

Was this also true for rural atmosphere in the NCP? By comparing with event days at
our site, we noticed that CS level was in general higher during non-event days. In other
words, NPF was very likely to occur when CS was significantly lowered. This strongly
demonstrates the similarity between our site with urban Beijing, that CS would be the
limiting factor for the occurrence of NPF. However, we noticed that besides the higher
CS, the H2SO4 concentration was also in general lower during non-event days
compared to event days. Particularly, there were a very few cases that CS was
somewhat quite low (<0.06 /s), being quite close to that under those event days, yet
NPF still did not occur, most likely owing to the lowered H2SO4 concentration at these
days. This implies that nucleating species, such as H2SO4, may not be always enough
to initiate nucleation at this site compared to urban Beijing, where pollution level were
typically higher. Under a certain CS, the level of H2SO4 relies on both the solar
radiation reaching to the Earth's surface and the concentration of gaseous SO2. By
looking at both content during non-event days (Fig. S2), we found that the
concentration of SO2 was comparable to that at Beijing and even far beyond that in
many clean environments that needed to initiate nucleation (for example, around 1
ppb in Hyytiälä). This means that the reduced solar radiation intensity was the main
reason determining the lower level of H2SO4 and thus controlling atmospheric NPF
not occur.

Taking both, we conclude that CS or CoagS was the dominating factor that governing
the occurrence and intensity of NPF at our GC site, while the influence from the
available H2SO4 is not negligible, either. The available H2SO4 was found to be mainly
determined by the solar radiation intensity reaching to the Earth but not SO2 level in
the atmosphere. This would be a general characteristic of NPF events in this region





compared to urban atmosphere of China, where threshold for CS was solely needed
for the appearance of NPF. Nevertheless, this pattern is also different from other
western countries, where the intensity of nucleating or condensable species is the
limiting and perhaps the only factor for the occurrence of NPF (Wang et al., 2017;
Kulmama et al., 2016; Kerminen et al., 2018).


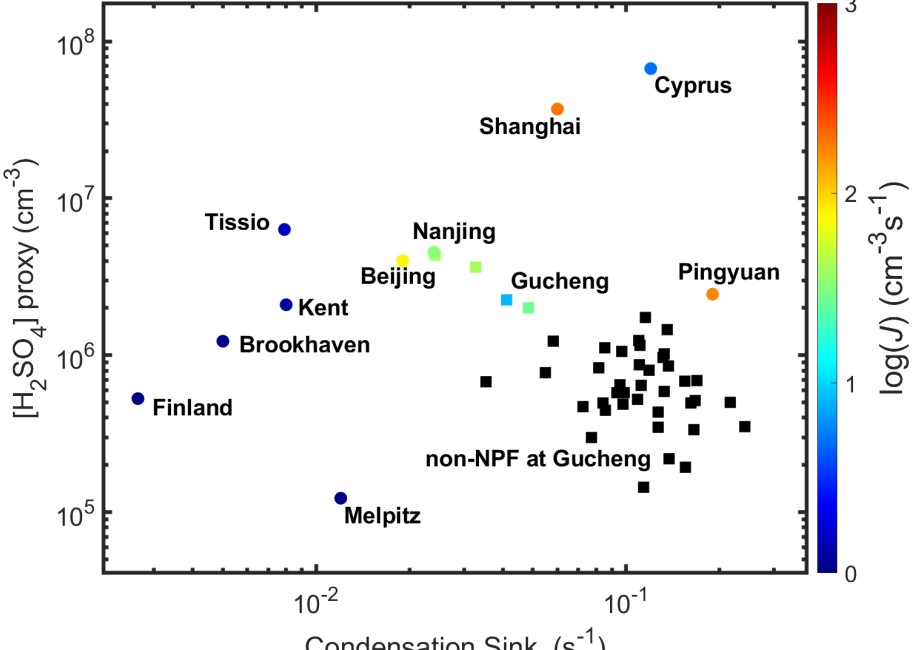


**Figure 5.** Condensation sink and $H_2SO_4$ concentration in various atmospheric environments,
including Finland (Nieminen et al., 2014), Tissio (Kalkavouras et al., 2020), Cyprus (Baalbaki et al.,
2020), Melpitz (Hamed et al., 2010), Brookhaven (Yu et al., 2014), Kent (Yu et al., 2014), Pingyuan
(Fang et al., 2020), Beijing (Deng et al., 2020), Nanjing (Herrmann et al., 2014) , and Shanghai (Yao
et al., 2018) site. The color bar indicates particle formation rate.

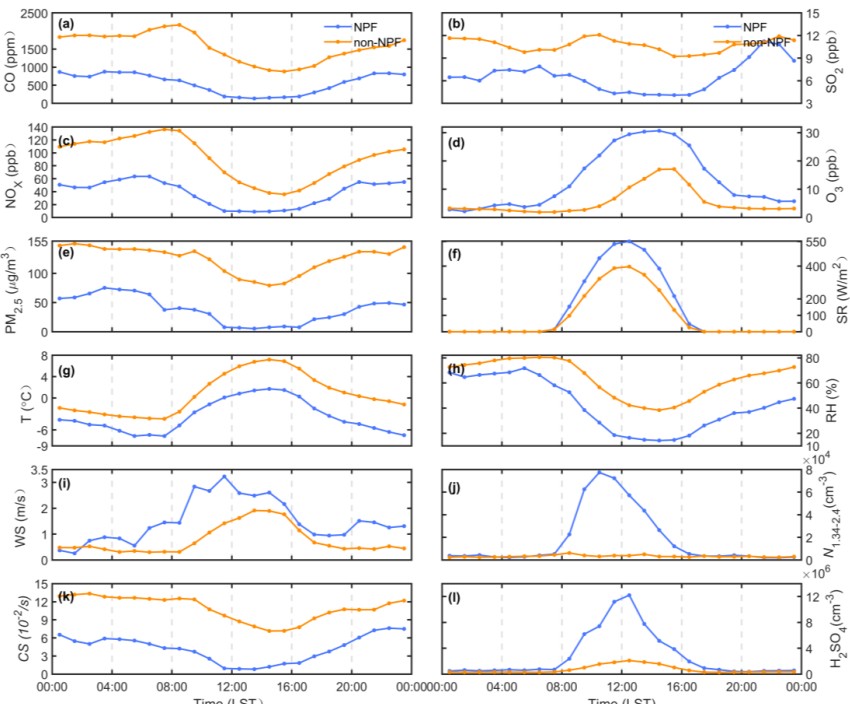


**Figure 6.** Diurnal variation of (a) CO, (b) $SO_2$, (c) $NO_x$, (d) $O_3$, (e) $PM_{2.5}$, (f) Solar radiation (SR), (g) *T*,
(h) RH, (i) wind speed (WS), (j) number concentration of sub-3nm cluster, (k) *CS*, and (l) $H_2SO_4$ proxy
during the NPF and non-NPF days during this field campaign.


On the other hand, we found that RH level under event days was generally lower than

that on non-event days (see Fig. 6). This is similar to the cases that NPF was observed

in Beijing by Yue et al. (2009), who suggested that photochemical reactions were faster

on sunny days with low RH. In addition to this, ambient temperature during NPF was

relatively lower than that on non-event days. Yan et al. (2021) considered that

temperature can affect the stability of H2SO4 clustering and thus influence NPF.

Therefore, all these factors could be the potential reasons increase or decrease the

probability of NPF to occur in current rural areas. It has to be noted that all these

features, including reduced RH level as well as ambient *T* during event days, could be

coincidence with reduced CS over clean days, for instance, being a consequence of air

masses originating from the north and bringing dryer, colder and cleaner air to the site.


Therefore, current discussion in this regard becomes ambiguous and may be inclusive,
but should still be considered separately when larger datasets are available. Moreover,
we observed that $O_3$ concentration was clearly higher during event days, implying that
other condensable vapours, for instance, organics, that involving $O_3$ oxidation, might
also be important to NPF in this region. Although these oxygenated organic
compounds may not necessarily participate in H2SO4 clustering, they may
considerably contribute to the growth of newly-formed particles, which should not be
ruled out in the study of NPF for this region and also need to be investigated in the
future.

4. Summary and conclusions

Most previous studies dealing with NPF in China were mainly based on measurements
of particles at larger sizes, typically above 3 nm, whereas detection of particles at sub-
3 nm range was quite limited. In our study, by coupling a PSM with a traditional SMPS,
We were able to measure the particle number size distribution down to 1.34 nm during
NPF events in the wintertime at a rural site of the NCP. Correspondingly, formation rate
of particles at 1.34 nm was obtained, widening the data pool concerning the feature
of NPF for this region. At current rural environment, high level of H2SO4 or low
concentration of fine particles may not always initiate the occurrence of NPF. Only at
the condition that the concentration of H2SO4 was relatively high and CS was
considerably low, NPF events were more likely to take place. This feature is slightly
different from that of the urban atmosphere of NCP, whereas NPF events were usually
characterized with high formation rate, high CS and high H2SO4 concentration.
However, as our H2SO4 concentration was predicted from empirical parameters,
particular cautions regarding their associated uncertainties should be considered. At
urban Beijing, NPF was also observed during the wintertime of 2018. We found that
their measured H2SO4 concentration was quite comparable to the predicted ones in
our study, indicating its relative reliability in using them though absolute uncertainties
could not be derived here. Yang et al. (2021) demonstrated that the derived fitting



parameters for the calculations of $H_2SO_4$ proxy may vary from site to site and between
different seasons. For instance, they considered the products from the ozonolysis of
alkenes were able to oxidize $SO_2$ to form gaseous $H_2SO_4$. Moreover, they pointed out
that $H_2SO_4$ could be from primary emissions, such as vehicles, or freshly emitted
plumes, which could account for 10% of the total $H_2SO_4$ in the atmosphere. These
aspects were not comprehensively considered in our calculations, which could bring
huge uncertainties or errors to the estimation. Thereby, though the $H_2SO_4$ proxy was
approximated to the measured ones at Beijing site, direct measurements for the $H_2SO_4$
concentration should be implemented in the future before driving any further
conclusion.






















**Declaration of interest statement.**

The authors declare that they have no known competing financial interests or personal relationships that could have appeared to influence the work reported in this paper.

**Data availability.**

The details data can be obtained from https://doi.org/10.5281/zenodo.7326388 (Hong, 2022).

**Author contributions.**

JH collected the resources, wrote and finalized the manuscript, MT analyzed the data, plotted the figures and wrote the original draft, QQW and NM planned the study, collected the resources, reviewed the manuscript. SWZ, SBZ, XHP, LHX, GL, UK conducted the measurements, CY, JCT, YK, YH, YQZ, WYX, GSZ, BY, ZBW discussed the results. YFC and HS contributed to fund acquisition.

**Competing interests.**

Hang Su and Yafang Cheng are members of the editorial board of Atmospheric Chemistry and Physics

**Acknowledgements.**

This work is supported by the National Natural Science Foundation of China (grant no. 42175117, 41907182, 41877303, 91644218) and the National key R&D Program of China (2018YFC0213901), the Fundamental Research Funds for the Central Universities (21621105), the Guangdong Innovative and Entrepreneurial Research Team Program (Research team on atmospheric environmental roles and effects of carbonaceous species: 2016ZT06N263), and Special Fund Project for Science and Technology Innovation Strategy of Guangdong Province (2019B121205004).



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
