# Peer review of "Measurement Report: Wintertime new particle formation in the rural area of North"

_Atmospheric Chemistry and Physics, 2022_

## Author Comment (AC1)

**Response to reviewer #1**

We appreciated referee#1's positive feedback and detailed suggestions which are very important for improving the quality of our manuscript. Our point-to-point replies to the referee's comments are listed below.

The measurement report "Wintertime new particle formation in the rural area of North China Plain" by Hong et al. presents measurements of particle size distribution at a rural measurement site, Gucheng, located at the North China Plain (NCP). The measurements were carried out between November and December 2018. With their measurements down to sub 3 nm particles, they aim to gain a better understanding of the new particle formation mechanisms in the rural area of the NCP and compare the results to studies in nearby urban environment.

The study shows deficiencies regarding citations, phrasing and data transparency, which have to be addressed.

Response: Thanks very much for the reviewer's comments. We went through the whole manuscript, rephrased relevant sections as suggested below, added the needed citations and clarified the used data more clearly.

General comments:

**1.** The authors refer to size bins of the PSM in nm size with 2 digit precision (e.g. line 1471.34-1.39 nm) and further refer to J rates and GR for 1.34 nm particles. I would suggest to round this number, as other studies show, that there is e.g. an uncertainty of 0.1 nm for the calibration of the PSM (Kangasluoma et al., 2015) and also other particle instruments, such as CPCs have an uncertainty for the cut-off size of 0.2 nm (e.g. stated in Dada et al., 2020). I am not sure the PSM can distinguish the sizes down to 0.01 nm difference in diameter.

Response: Thanks very much the specific comment. Yes, we agree that reporting those values with 2 digit precision is improper. Therefore, we rephrased all the relevant parts with only 1 digit precision.

**2.** In section 2.2.3 the authors mention measurements with an ACSM, however I did not find any data shown from this instrument in this measurement report. Was the data used for the study? If so, could it support the NPF mechanism assumptions of the study?

Response: To support the NPF mechanism assumptions of current study, chemical composition of nanoparticles below 50 nm, particularly below 20-30 nm might be quite useful. However, as the ACSM measured the bulk chemical composition of aerosols, which deviates significantly from that of small particles, it might not be a good asset in supporting these discussions. We thereby removed the sentence in section 2.2.3 referring ACSM measurements.

**3.** Do I interpret this correctly from Fig. 2 that there are NPF events at the GC site almost every day, but they seem to be transported? And on certain days, such as the example in Fig. 3, there is NPF in the range of 1.3-3.7 nm and further a growth is visible in the larger particles, measured by SMPS? Could the authors be more precise regarding the NPF event classification? In Fig. 2 it seems like most events are transported events, since there are barely any paritcles below approx. 18 nm.

Response: We understand the reviewer's doubt that from Fig.2 in the original manuscript there seems NPF events occurring at current site almost every day. Figure 1 below shows the time series of particle number size distribution in the size range of 10-450 nm measured by a nano-SMPS on November 19, 2018. This figure indicates a clear example that plenty of nucleation-mode particles with size above 10 nm did appear during some days of our measurements. However, we did not observe the burst of sub-3 nm clusters from the PSM measurements and moreover no clear growth of these particles can be identified. On the one hand, it demonstrates that these particles probably are not from nucleation of $H_2SO_4$ with other species and their subsequent growth, but more probably local emissions (traffic exhausts) or transported. On the other hand, it also hinders an accurate calculation of the particle formation and growth rate if being classified as NPF days. Therefore, these cases, either from local emissions or transported with substantial nucleation mode particles, were classified as non-event days in our study and only the ones with a burst of sub-3 nm particles and a typical banana-shape in PNSD were considered as NPF events, of which the potential mechanism and influencing factors were discussed further. Accordingly, we rephrased the part in Sect. 2.3.4 to be more precise.

On line 200-206, we revised the section as:
"*Days of NPF events was classified according to the method proposed by Dal Maso et al. (2005) and Kulmala et al. (2012), in which (a) a burst in the concentration of sub-3 nm particles or clusters was observed and (b) these particles had a continuous growth over a time span of hours (e.g., usually more than ten hours). If no clear growth of these newly formed particles (sub-3 nm particles) can be identified, the day was classified as an undefined day. The day without both the burst of sub-3 nm particles and their subsequent growth was considered as a non-event day.*"

Furthermore, we revised the discussion on 219-227 by including the description of these undefined and non-event days as:
"*According to the PNSD and PSM data, five days, with four of which having significant burst of sub-3 nm clusters as shown in Fig.2e, were classified as NPF events out of the total experimental period. It has to be noted that on the day of November 18, though PSM data was not available due to technical issues, clear growth of nucleation mode particles with a typical banana-shape PNSD was observed, lasting for more than 12 hours. These particles under the growth of such a long time should not be from traffic emissions or transported. Therefore, it was also classified as an event day in our study. Considering all these five NPF event, this corresponds to an NPF frequency of 12.8%, which was lower compared to those at an urban site (i.e., Beijing) in the same region during the same season (Shen et al. (2018) (25.8%); Deng et al. (2020) (51.4%)). Similar findings were also observed in Yue et al. (2009) and Wang et al. (2013), that NPF frequencies were higher at the Beijing urban site than at the corresponding regional background or rural site. Yue et al. (2009) and Wang et al. (2013) attributed this to the higher pollution level and correspondingly higher precursor content in the urban cities, leading to stronger NPF events there.*
*During our study, six days, with a slightly weak burst of sub-3 nm particles, were identified as undefined days as their formation and growth rate cannot be calculated accurately. For non-event days, we observed that during many of them some nucleation-mode particles with size above 10 nm did appear. However, we did not observe the burst of sub-3 nm clusters from the PSM*

*measurements and moreover no clear growth of these particles can be identified. This indicates that these small particles probably are not from nucleation of H₂SO₄ with other species and their subsequent growth, but more likely local emissions (traffic exhausts) or long-range transported.*"

[Figure]

Figure 1: Particle number size distribution in the size range of 10-450 nm measured by a nano-SMPS on the date of November 19, 2018.

**4.** The authors mention several measurements of tracehouse gases, the proxy of sulfuric acid and ACSM data, however only show a fraction or mean of it. I would suggest to add those measurements in the time series plot of figure 2 or potentially in the supplementary, so that the reader gets a clearer idea of the behaviour of the various parameters.

Response: Thanks for the comment. Time series of the concentration of trace gases, including CO, O₃, NO, NO₂, SO₂ were added in the supplement material. We also show this figure here for clarification. Time series of the concentration of the derived H₂SO₄ were added in Fig.2 in the revised manuscript. Corresponding discussion regarding the concentration of these trace gases was also added in the revised manuscript on line 217 as:

"*The observed time series of concentration of different trace gases during current study is shown in Fig. S1. To be specific, the campaign-averaged concentration of CO, O₃, NO$_X$ and SO₂ was 1394 ppb, 7 ppb, 83 ppb and 10 ppb, respectively.*"

[Figure]

Figure 2: Time series of the measured concentration of different trace gases e.g., CO, O₃, NO, NO₂, NO$_X$, SO₂ during current campaign.

**5.** Figure 5 is not mentioned anywhere in the text. Please include or remove the figure if irrelevant for the study.

Response: Thanks for the comments. As SA was re-calculated according to a new method, the results in this section were all revised. We replaced Fig. 5 in the original manuscript with a new figure (shown below). And we introduced a new parameter (*I*) here to discuss the influencing factors controlling the occurrence of NPF events. The description of *I* (we added a new section in the data processing part) and corresponding results and discussions were added in the revised manuscript.

After line 206:

"*Sect 2.3.5 Indicator for the occurrence of NPF*

*Previously, McMurry et al. (2005) proposed a dimensionless criterion, L, to predict the occurrence of NPF events in the atmosphere. After being validated in diverse atmospheric environments (Kuang et al., 2005; Cai et al., 2017; ), L has been used to investigate the governing factors for NPF events under typical atmospheric conditions. Upon recently, Cai et al. (2021) proposed a new indicator, I, on the basis of L, which only considered $H_2SO_4$ to drive the growth. The new indicator was calculated by further taking into account the condensation of other species, for instance, amines and has been suggested to be a good quantitative representation for the occurrence of NPF after comparing with L for NPF events observed at urban Beijing (Deng et al., 2020). The detailed information to calculated I can be found in Cai et al (2021).*"

Line 361-390:

"*Was this also true for rural atmosphere in the NCP? By comparing with non-event days at our site (see Fig. 5a), we noticed that H2SO4 level was not significantly higher but sometimes even lower than that during non-event days. In other words, the abundance of $H_2SO_4$ did not always lead to NPF; and it was only when CS was significantly lowered that the event became more likely to occur. This strongly demonstrates the similarity between our site with urban Beijing, that CS would be the limiting factor for the occurrence of NPF. However, we noticed that there were a very few cases (two cases) that CS was somewhat quite low, being quite close to that under those event days, yet NPF still did not occur. The most plausible explanation for this could be on the one hand the lowered $H_2SO_4$ concentration at these days (as shown in Fig. 5a) and on the other hand the other nucleating species rather than $H_2SO_4$ may not be always enough to initiate nucleation at this site.*

*As previously stated that the dimensionless criterion, I, is a good quantitative indicator to predict whether an NPF occurs or not during a certain day, we plotted I against the condensational sink for NPF days and other days under different $H_2SO_4$ level. Cai et al. (2021) found that the larger the I value, the higher frequency that NPF events occurred for both urban Beijing and Shanghai, which was also clearly revealed by our results. On the one hand, as shown in Fig. 5b, the largest I values were mostly observed for NPF days, confirming its feasibility in predicting the occurrence of NPF events. On the other hand, the obtained I anti-correlated with CS quite well, while the influence from the available $H_2SO_4$ was not obvious. This strongly suggests that CS was the dominating factor governing the appearance of NPF events at current environment, being highly consistent with the feature in Beijing.*"

[Figure]

Figure 3: (a) H₂SO₄ concentration as a function of condensation sink during both event days (squares) and no-event days (circular dots) during our study. (b) The dimensionless indicator, I, as a function of condensation sink. The colorbar indicates: solar radiation (left panel) and H₂SO₄ proxy concentration (right panel).

Specific remarks:

Line 41: "This implies that H2SO4-amine nucleation..." What is meant by "This"?, the fact that both locations show NPF at the same time? Did the air mass origin from the urban site? Please rephrase to be more clear.

Response: Thanks for the comment. "*This*" means "*the simultaneous occurrence of NPF events in both places*". We calculated the 72-h back trajectories of air masses arriving at our GC site and urban Beijing at 6-h intervals using the HYbrid Single-Particle Lagrangian Integrated Trajectory (HYSPLIT) model. We found that the transport paths of the air masses arriving at our site was roughly similar to that of urban Beijing, both originating from Siberia area where concentration of gaseous pollutants and particulate matter was typically quite low, which is shown as an example on the date of 7 Dec 2018 in Fig. 4 below. We added some description of HYSPLIT analysis in the end of Sect. 2.2.3 as:

"*Furthermore, in order to investigate the influence of the origins and transport paths of air parcels to the local atmospheric compositions during NPF events, 72-h back trajectories of air masses arriving at 100 m above ground level at our GC site were analyzed using the HYbrid Single-Particle Lagrangian Integrated Trajectory (HYSPLIT) model for the classified event days.*"

Moreover, we added more discussion including these analysis into Sect. 3.2 on line 314 to support our conclusions.

"*Additionally, we found that during these five NPF events air masses arriving at our site followed similar transport paths to that at urban Beijing (see Fig. S2 as an example in the supplement), both originating from Siberia area through the northwest of observational site, where concentration of gaseous pollutants and particulate matter was typically quite low. Taking both evidence, we hypothesis that NPF events during these days in this region might be a regional phenomenon, sharing the same or similar nucleation mechanism.*"

We also agree with the reviewer that the sentence on line 41 is not proper and thus we modified it into:

"*During these five days, NPF concurrently occurred in an urban site in Beijing. Sharing similar sources and transport paths of air masses arriving at our site to that of urban Beijing, we hypothesis that NPF events during these days in this region might be a regional phenomenon. The simultaneous occurrence of NPF in both places implies that $H_2SO_4$-amine nucleation, concluded for urban Beijing there, could also be the dominating mechanism for NPF at our rural site.*"

[Figure]

Figure 4: 72-hour back trajectories of air masses arriving at our GC site as well as urban Beijing on December 7, 2018

Line 42-46: This sentence sounds ambigous; the CS and CoagS are factors that potentially limit the nucleation, as they act as a sink, while H2SO4 is a precursor. Those are two different things, affecting NPF from different angles. Please rephrase the sentence to correct this statement.

Response: Thanks for the comments, we rephrased the sentence as:

"*The higher concentration of sulfuric acid during many non-event days compared to that of event days indicates that the content of sulfuric acid may not necessarily lead to NPF events under current atmosphere. Only when the condensation sink or coagulation sink was significantly lowered, atmospheric NPF occurred, implying that CS or CoagS are the dominating factor controlling the occurrence of NPF for present rural environment of NCP.*"

Line 61: "H2SO4 are the key precursors": Sulfuric acid is one of the key precursors but not the only one as stated here. Other examples are iodic acid (He et al., 2021, Sipilä et al., 2016) or highly oxygenated organic molecules (Kirkby et al., 2016).

Response: Thanks for the comment. We discussed the role of other precursors in NPF the later part of that paragraph. We agree that current sentence was improper and thus modified it into:

"*Numerous laboratory measurements and field studies have shown that sulfuric acid molecules ($H_2SO_4$) are one of the key precursors to form molecular clusters for nucleation (Nieminen et al., 2010; Sipilä et al., 2010; Kirkby et al., 2011; Riccobono et al., 2014; Stolzenburg et al., 2020).*"

Line 70-71: please cite original publications describing the NAIS rather than publications including the instrumentation only in the methods section (Mirme & Mirme, 2013)

Response: Thanks for the comment, we cited the original publications (Mirme & Mirme, 2013) describing the NAIS in the revised manuscript accordingly.

Line 71: same as above, include citations which actually describe the instrument when introducing them (Jokinen et al., 2012)

Response: Thank you for your comment, we cited the article (Jokinen et al., 2012) in the revised manuscript.

Line 143: see above, please include original description of the PSM as citation (Vanhanen et al., 2011)

Response: Thanks, we included the original description (Vanhanen et al., 2011) of the PSM as citation in the revised manuscript.

Line 153: Typo in the name of the instrument: Thermo Fisher Analysers

Response: Thank you for your comment, we changed "*Themo Fisher Analyzers*" into: "*Thermo Fisher Analysers*".

Figure 2: I would suggest to change the y-axis limit and label of the PNSD (panel d) to 10 nm as stated in the text and figure caption

Response: Thank you for your comment, we changed the y-axis limit and label of the PNSD accordingly. This figure was also shown below for clarification.

[Figure]

Figure 5: Time series of (a) wind speed and wind direction, (b) temperature (*T*) and relative humidity (RH), (c) total particle surface and volume concentration calculated by using PNSD data, (d) measured PNSD in the size range of 10 - 800 nm, (e) particle number concentration in the range of 1.3 to 2.4 nm and $H_2SO_4$ proxy concentration during the entire measurement period (2018.11.12-2018.12.24). White portion indicates no data was available due to instrument maintenance or power failure. Note that white portion in the PNSD in the size range of 10 - 15 nm, indicating no available data, is due to the technical problems of our SMPS system; therefore data for that time period from a parallel SMPS covering sizes of 15 - 800 nm was used instead.

Figure 3: the y-label should show that it is a H2SO4 proxy concentration

Response: Thanks for your comments. We modified the figure label in the revised manuscript, as shown below.

[Figure]

Figure 6: A case of NPF event on December 7 during this field campaign. Time series of (a) wind speed and wind directions, (b) the PNSD in the size range of 10 - 450 nm (The white dotted line represents the size with diameter at 25, 50, and 100 nm; black line represents the polynomial fit of the measured PNSD, (c) the particle number concentration of nucleation mode (9 - 25nm) and CS, (d) the number concentration of sub-3nm clusters and predicted concentration of sulfuric acid.

Line 255-257: Please rephrase the sentence, the meaning is unclear.

Response: We modified the sentence as:

"*Note that most atmospheric formation rates reported in China were based on the measured formation rates at relatively larger size, i.e., 3-10 nm, which are so called the "apparent" particle formation rates. In order to derive the formation rates of critical clusters from the "apparent" particle formation rates (Kulmala et al., 2017), the nuclei GR or GR at sub-3 nm is needed but usually remains unclear.*"

Line 267: "clearly proved" (correct would be: clearly proven) sounds too certain. I would be more cautious with the statement and suggest something like "is indicated by"

Response: Thank you for your comments. We modified the sentence into:

*"The most plausible explanation could be the higher abundance of nucleating precursors for NPF in those polluted atmosphere, which is indicated by the SA concentration, either measured in urban Shanghai and Nanjing or calculated in our study."*

Line 268: I suggest to add the information here, that it is an SA proxy concentration

Response: We added the information in the revised manuscript as:

*"To be specific, the mean SA proxy concentration during NPF at our GC site was around $1.4 \cdot 10^7$ $cm^{-3}$ , a factor of around 30 higher than that at Hyytiälä in Finland."*

Line 267-269: I suggest to add the SA proxy concentration in Fig. 2 so that the reader can see the daily variation, and to support the mean SA concentration mentioned here at GC measurement site.

Response: Thank you for your comments. We added the SA proxy concentration in Fig. 2 of the revised manuscript, which is also shown previously.

Line 269-270: please add a citation for the Hyytiälä, Shanghai and Nanjing SA concentration

Response: Yes, we added the citations for the Hyytiälä (Nieminen et al., 2014), Shanghai (Xiao et al., 2015) and Nanjing (Herrmann et al., 2014) SA concentration in the revised manuscript.

Line 287 -288: add citations for the GR at Hyytiälä, Jungfraujoch, Beijing and Pingyuan

Response: Thanks for your comments. We added the citations for the reported GR at Hyytiälä (Kulmala, 2013), Jungfraujoch (Boulon et al., 2010), Beijing (Chu et al., 2021) and Pingyuan (Fang et al., 2020) in the revised manuscript.

Line 325-329: What is the meaning of this sentence? Was the I-APi-TOF deployed at GC measurement site? It was not mentioned in the methods, only a ACSM. Please clarify.

Response: Thanks for your comments. We added the description of I-APi-TOF measurement in the method part as:

*"The concentration of oxygenated volatile organic compounds (OVOCs) was measured with an iodide-adduct long time-of-flight chemical ionization mass spectrometer (I-CIMS, Aerodyne, US) at a time resolution of 10-30 s for current study."*

Line 349: here you mention the concentration of VOCs. In the method section I could not find any information regarding VOC measurements. Please clarify.

Response: This sentence actually describes a general condition of the NCP of China compared to many other environments. Therefore, we added a reference (Chu et al., 2019) here to clarify.

Line 367-370: The sentence is very confusing, please rephrase.

Response: We revised the whole section as:

*"Was this also true for rural atmosphere in the NCP? By comparing with non-event days at our site (see Fig. 5a), we noticed that $H_2SO_4$ level was not significantly higher but sometimes even lower than that during non-event days. In other words, the abundance of $H_2SO_4$ did not always lead to NPF; and it was only when CS was significantly lowered that the event became more likely to occur. This strongly demonstrates the similarity between our site with urban Beijing, that CS*

*would be the limiting factor for the occurrence of NPF. However, we noticed that there were a very few cases (two cases) that CS was somewhat quite low, being quite close to that under those event days, yet NPF still did not occur. The most plausible explanation for this could be on the one hand the lowered $H_2SO_4$ concentration at these days (as shown in Fig. 5a) and on the other hand the other nucleating species rather than $H_2SO_4$ may not be always enough to initiate nucleation at this site.*

*As previously stated that the dimensionless criterion, I, is a good quantitative indicator to predict whether an NPF occurs or not during a certain day, we plotted I against the condensational sink for NPF days and other days under different $H_2SO_4$ level. Cai et al. (2021) found that the larger the I value, the higher frequency that NPF events occurred for both urban Beijing and Shanghai, which was also clearly revealed by our results. On the one hand, as shown in Fig. 5b, the largest I values were mostly observed for NPF days, confirming its feasibility in predicting the occurrence of NPF events. On the other hand, the obtained I anti-correlated with CS quite well, while the influence from the available $H_2SO_4$ was not obvious. This strongly suggests that CS was the dominating factor governing the appearance of NPF events at current environment, being highly consistent with the feature in Beijing.*"

Line 375: Add citation for SO2 at Beijing
Response: Thanks for the comments. We did not use SO2 data at Beijing anymore here.

Line 376-377: add citation for SO2 at Hyytiälä
Response: Thanks for the comments. We did not use SO2 data at Hyytiälä anymore here.

Line 377-379: This statement is quite daring, as it might also be that other precursors were simply not abundant to initiate NPF. As previously mentioned (line 370), the NPF mechanism at the measurement site is probably a mixture of several precursors.
Response: Thank you for your comments. We modified the whole section discussing the governing factors for the occurrence of NPF in the rural environment of NCP in China.

Line 385-387: Please add a citation for this statement
Response: We removed this statement here in the revised manuscript due to new discussion.

Line 387-390: It is not clear to me, how this is differnt. It was stated that the H2SO4 concentration is mainly depending on the solar radiation, thus the concentration of H2SO4 as precursor is determining (among probably other precursors) whether there is NPF or not. Now it is stated that this is different from other countries where the "intensity" (I assume this means concentration) determines the NPF occurence. Please clarify.
Response: Thank you for your comments. We modified the whole section discussing the governing factors for the occurrence of NPF in the rural environment of NCP in China. Current sentence was removed.

Figure 6: Is the diurnal variation calculated for all NPF and non-NPF days? I suggest to mention how many days are included in the mean calculation to be more transparent with the overview and how much one can interpret from these figures.

Response: Yes, the diurnal variation was calculated for all NPF and non-NPF days. These values were averaged over the five NPF days and 28 non-event days, respectively. We added this information into the figure caption.

Line 409-410: A brief comment: Generally lower temperatures favour nucleation, also citations from the previously mentioned CLOUD experiments regarding RH and temperature could be included here.
Response: Thanks for the comments. We added the citations (Kirkby et al. (2011); Riccobono et al. (2014)) here.

Line 419: I would add to the sentence "that involve, among others, O3 oxidation", as also OH, NO3 are oxidants involved in forming HOM.
Response: Thanks for the comment. We modified the sentence into:
"*Moreover, we observed that $O_3$ concentration was clearly higher during event days, implying that other condensable vapors, for instance, organics, that involve $O_3$, among others, in forming HOM, might also be important to NPF in this region.*"

Line 420: When referring to "these oxygenated organic compounds", they should be elaborated on more thoroughly and a citation should be included.
Response: Thanks for the comment. We modified the sentence as:
"*Although these organic compounds formed through $O_3$ oxidation may not necessarily participate in $H_2SO_4$ clustering, they may considerably contribute to the growth of newly-formed particles (Mohr et al., 2019), which should not be ruled out in the study of NPF for this region and also need to be investigated in the future.*"

Language remarks:

Line 41 "... might be a regional phenomena" phenomenon
Response: Thank you for your comment, we changed "*...might be a regional phenomena*" into "*...might be a regional phenomenon*".

Line 58 "CCNs" CCN
Response: Yes, we changed "*CCNs*" into "*CCN*".

Line 172: "being mainly for PSM data": a word is missing here
Response: Thanks for the comment. We modified the sentence into: "*being mainly for the PSM data*".

Line 185: "fresh formed" freshly formed
Response: Thank you for your comment, we changed "*fresh formed*" into "*freshly formed*".

Line 239: "dramatic" rather use something like "rapid"
Response: Thank you for your comment, we changed "*dramatic*" into "*rapid*".

Line 254: "formation rate [...] was based" should be plural: rates [...] were based

Response: Thanks for the comment. We modified the sentence into:

"*Note that most atmospheric formation rates reported in China were based on the measured formation rate at relatively larger size, i.e., 3-10 nm.*"

Line 265: "could be the more abundance" the higher abundance

Response: Thank you for your comment, we changed "*could be the more abundance*" into "*could be the higher abundance*".

Line 312: "during a much longer time" for a longer period of time

Response: Thank you for your comment, we changed "*during a much longer time*" into "*for a longer period of time*".

Line 315: "might be a regional phenomena" phenomenon

Response: We changed "*might be a regional phenomena*" into "*might be a regional phenomenon*".

Line 319: "our results deviates" deviate

Response: Thank you for your comment, we changed "*our results deviates*" into "*our results deviate*".

Line 333: "on the other side" on the other hand

Response: We changed "*on the other side*" into "*on the other hand*".

Line 335: "dominating one varies" varying

Response: We changed "*with the dominating one varies upon...*" into "*with the dominating one varying upon...*".

Line 381: "Taking both" a word is missing

Response: This section was revised with current sentence being removed.

Line 384: remove "reaching to the Earth"

Response: Thank you for your comment, we removed the whole sentence.

Line 390: I believe the citation should be Kulmala, the mistake repeats in the reference list

Response: Thank you for your comment, we revised the citation and reference list.

Line 419: "that involving O3 oxidation" that involve O3

Response: Thank you for your comment, we changed "*that involving O3 oxidation*" to "*that involve O$_3$*".

Line 439: "high formation rate" rates

Response: Thank you for your comment, we changed "*high formation rate*" into "*high formation rates*".

Line 448-449: this sentence lacks a word or punctuation mark

Response: Thank you for your comment, we revised the sentence into:

"*Moreover, they pointed out that $H_2SO_4$ could be from primary emissions, such as vehicles or freshly emitted plumes. The $H_2SO_4$ from these sources could account for 10% of the total $H_2SO_4$ in the atmosphere.*"

Supplementary material:

Figure S1: The figure caption states that the left panel is showing formation rate as a function of SA concentration, however the label shows SA*DIACIDS and the concentration is given in 10^17 - 10^18 range. It is not clear what is meant here, is it a multiplication product or clusters, and how is this concentration retrieved? It seems extremely high.

Response: Thanks for the comment. We removed this figure in the revised manuscript and did more data analysis regarding the SA-Diacid nucleation mechanism. New figures were plotted and the discussions were modified as well accordingly.

Figure S2: "square dots" should be denoted as squares, otherwise it is contradictory

Response: Thank you for your comment. This figure was removed in the revised supplement.

---

## Author Comment (AC2)

**Response to reviewer #2**

We appreciated referee#2's positive suggestions and constructive comments. Our point-to-point replies to the referee's comments are listed below.

Hong et al. presents a measurement report on wintertime new particle formation (NPF) in the rural area of North China Plain, exploring its influencing factors and possible formation mechanism. While NPF in urban cities of China are relatively well investigated, measurements in the rural areas are rather not enough to understand its underlying mechanisms. This manuscript is of great interest to readers of general aerosol formation, particularly in the NPF research. However, the following issues/concerns need to be addressed before it can be accepted for publication in the journal.

1. The sulfuric acid (SA) concentration in this study was calculated according to proxy proposed by Petaja et al. (2009). However, Lu et al. (2019) proposed a better proxy for estimating SA concentration in urban Beijing. It would be beneficial to compare both proxies for their performance since Petaja et al.'s proxy is based on measurements in boreal environments while Lu et al's one might be more applicable to the current study.

Response: We thank the reviewer for the suggestions and agree with the reviewer. We recalculated the SA concentration using the proxy proposed by Lu et al. (2019). All the corresponding figures, tables and discussion were modified accordingly.

2. This study measured only particle number size distribution and no potential nucleating precursors (e.g., sulfuric acid, amines, carboxylic acids) were measured. A SA-DMA mechanism was proposed based on the similarity of correlations of the particle formation rate with sulfuric acid concentration for this study and for the Cloud studies. It seems that proposing such a mechanism based solely on this similarity does not make any sense, providing no any measured DMA concentration in this area. In addition, how can the authors ensure the base is DMA rather than any other amines?

Response: We agree with the reviewer that without direct measurements of potential nucleating precursors, it might be improper to make such a strong conclusion regarding the nucleation mechanism. Therefore, we focused more on the comparison of the NPF between our study and that observed at urban Beijing. For example, we did more analysis by calculating the 72-hour back trajectories of air masses arriving at our site as well as at urban Beijing. We found that that the transport paths of the air masses arriving at our site was roughly similar to that of urban Beijing, both originating from Siberia area where concentration of gaseous pollutants and particulate matter was typically quite low, which is shown as an example on the date of 7 Dec 2018 in Fig. 1 below. Together with the fact that NPF concurrently occurred at urban Beijing during the same days, we hypothesis that NPF events during these days at current region might be a regional phenomenon, most likely sharing the same or similar nucleation mechanism. This further suggests that SA-DMA nucleation, which was confirmed for urban Beijing, could potentially be the dominating mechanism for GC site. We agree that our previous conclusion sounds quite arbitrary and we thereby rephrased our statement in this section, as shown below:

On line 297-305:
"*To further understand the dominating nucleation mechanism in the rural atmosphere of NCP in*

*China, we plotted the measured formation rate of 1.3 nm particles ($J_{1.3}$) against the simulated $H_2SO_4$ concentration and compared the results to previous studies conducted in different environments, as shown in Fig. 4. As illustrated by the significant correlation between the concentration of sulfuric acid and the particle formation rates, sulfuric acid is considered to be the driving species in the initial steps of NPF as confirmed conventionally. However, the obtained $J_{1.3}$-$H_2SO_4$ relationship for current environment appeared to deviate largely from those obtained by other studies. If only referring to the slope of the $J_{1.3}$-$H_2SO_4$ relationship, our results seem to approximate most to the ones measured by these CLOUD (The Cosmics Leaving OUtdoor Droplets chamber) experiments based on the mechanism of $H_2SO_4$-DMA nucleation. However, without the direct measurements of other potential precursors, the molecules stabilizing $H_2SO_4$ clustering still remain unclear.*"

On line 307-317:

"*Comparing the particle formation rates reported in different environments in China, our results were of the similar magnitude as that in Beijing (Cai et al., 2021), an urban site in the NCP. It has to be noted that their study was conducted during a much longer time and completely covered the measurement period of our study. More importantly, during the five days of events in our study, NPF concurrently occurred at their measurement site (Liu et al., 2020). Additionally, for these five event days air masses arriving at our site followed similar transport paths to that at urban Beijing, both originating from Siberia areas, where concentration of gaseous pollutants and particulate matter was typically quite low, through the northwest of the observational sites. Taking both evidence, we hypothesis that NPF events during these days in this area might be a regional phenomenon, sharing the same or similar nucleation mechanism. Cai et al. (2021) and Yan et al. (2021) further concluded that $H_2SO_4$-DMA was the dominating nucleation mechanism for urban Beijing with an additional support from the measured C2-amine concentration. Considering the similarities between these two sites, we speculated that the clustering of $H_2SO_4$ with DMA may also dominate the nucleation process at our site during winter, though future work is needed to verify current hypothesis.* "

[Figure]

Figure 1: 72-hour back trajectories of air masses arriving at our GC site as well as urban Beijing on December 7, 2018.

3. The exclusion of SA-acid mechanism was based on Figure S1 which show a bad correlation between J and SA*diacids. However, the rationale needs to be validated since concentrations of both SA and diacids varied in the correlation plot, leading to violation of thermodynamic roles. This approach is not convincing.

Response: Thanks for the comments. We agree with the reviewer and thus did more data analysis regarding the temporal pattern of these precursors concentration instead of only looking at the correlation plot. As illustrated in Fig. 2 below, the concentration of these four dicarboxylic acids during NPF events were in general lower than that during non-event days, on which sulfuric acid was also abundant that clustering of these two precursors should be more favored but NPF still did not occur. By further looking at the diurnal trend of the level of these precursors in the atmosphere for NPF days (shown in Fig. 3), we found that the rise in the SA concentration during day time of NPF events was not concurrently observed with or followed by the elevation of the diacids content. On the contrary, the signals of these diacids normally peaked during nighttime with the lowest level around daytime when NPF was typically initiated. Together with these evidence, we hypothesis that $H_2SO_4$-diacid may not be the dominating mechanism for the NPF of current environment. We rephrased the discussion in this section accordingly.

On line 325-336:
"*However, as illustrated in Fig. S4, the concentration of these four dicarboxylic acids during NPF events were in general lower than that during non-event days. Furthermore, during the daytime of events days when NPF was typically initiated, the signals of these diacids obtained from the I-CIMS did not show clear increase, unlike sulfuric acid, but rather elevated during the night time (see Fig. S5), being obviously different from the case of Pingyuan. Hence, the involvements of diacids during the initial steps of nucleation under current rural atmosphere might not hold. This statement does not necessarily mean that our previous inference was incorrect, but on the other hand, provides some hints that though NPF events in the NCP is regional, there might be no uniform theory but multiple mechanisms coexisting to explain its feature with the dominating one varying upon different emission patterns or meteorological conditions.*"

[Figure]

Figure 2: Measured concentration of these four dicarboxylic acids as well as the concentration of SA proxy during our study.

[Figure]

Figure 3: Diurnal trend of the concentration of SA as well as these dicarboxylic acids in the atmosphere for NPF days.

4. The authors conclude that the controlling factor for NPF is the condensational sink which is based on qualitative comparison of CS characteristics between the measurement site and urban Beijing. However, some quantitative representations are needed to make sure the CS indeed is the most important factor for determining the occurrence of NPF at the site.

Response: Thanks for the comments. Here, we used a dimensionless criterion, *I*, proposed by Cai et al. (2021) to quantitatively represent our conclusion that the controlling factor for NPF is the condensational sink. The description of *I* (we added a new section in the data processing part) and corresponding results and discussions were added in the revised manuscript. The calculated *I* as a function of the condensational sink was also shown here for clarification.

[Figure]

Figure 4: The demensionless criterion, *I*, as a function of CS. The colorbar indicates the concentration of $H_2SO_4$. The squares indicate NPF days while the circles are for non-event days.

After line 206:

*"Sect 2.3.5 Indicator for the occurrence of NPF*

*Previously, McMurry et al. (2005) proposed a dimensionless criterion, L, to predict the occurrence of NPF events in the atmosphere. After being validated in diverse atmospheric environments (Kuang et al., 2005; Cai et al., 2017; ), L has been used to investigate the governing factors for NPF events under typical atmospheric conditions. Upon recently, Cai et al. (2021) proposed a new indicator, I, on the basis of L, which only considered $H_2SO_4$ to drive the growth. The new indicator was calculated by further taking into account the condensation of other species, for instance, amines and has been suggested to be a good quantitative representation for the occurrence of NPF after comparing with L for NPF events observed at urban Beijing (Deng et al., 2020). The detailed information to calculated I can be found in Cai et al (2021)."*

Line 361-390:

*"Was this also true for rural atmosphere in the NCP? By comparing with non-event days at our site (see Fig. 5a), we noticed that $H_2SO_4$ level was not significantly higher but sometimes even lower than that during non-event days. In other words, the abundance of $H_2SO_4$ did not always lead to NPF; and it was only when CS was significantly lowered that the event became more likely to occur. This strongly demonstrates the similarity between our site with urban Beijing, that CS would be the limiting factor for the occurrence of NPF. However, we noticed that there were a very few cases (two cases) that CS was somewhat quite low, being quite close to that under those event days, yet NPF still did not occur. The most plausible explanation for this could be on the one hand the lowered $H_2SO_4$ concentration at these days (as shown in Fig. 5a) and on the other hand the other nucleating species rather than $H_2SO_4$ may not be always enough to initiate nucleation at this site.*

*As previously stated that the dimensionless criterion, I, is a good quantitative indicator to predict whether an NPF occurs or not during a certain day, we plotted I against the condensational sink for NPF days and other days under different $H_2SO_4$ level. Cai et al. (2021) found that the larger the I value, the higher frequency that NPF events occurred for both urban Beijing and Shanghai, which was also clearly revealed by our results. On the one hand, as shown in Fig. 5b, the largest I values were mostly observed for NPF days, confirming its feasibility in predicting the occurrence of NPF events. On the other hand, the obtained I anti-correlated with CS quite well, while the influence from the available $H_2SO_4$ was not obvious. This strongly suggests that CS was the dominating factor governing the appearance of NPF events at current environment, being highly consistent with the feature in Beijing."*

Below are rather minor:

- There are lots of typos, ill-sentences through the manuscript which need to be corrected.
Response: Thanks for the comments, we went through the manuscript and corrected all relevant ill-sentences.

- L39, at an urban site
Response: Thanks for the comments, we changed to *"at an urban site"*.

- L61 and throughout the text, H2SO4 needs subscript for 2 and 4. There are lots of such typos in the text for H2SO4 and other molecular formula

Response: Thanks for the comments, we went through the manuscript and corrected all typos relevant to subscription.

- L61, sulfuric acid molecules, follow a plural form

Response: We added "*molecules*" after "*sulfuric acid*".

- L67, newly form

Response: We changed to "*newly formed*".

- L73-74, I don't think the chemical composition of 1-3 nm particles has been speciated thus far.

Response: Thanks, we agree and we deleted "*as well as the chemical composition*" in the sentence.

- L80, important contributors to atmospheric nucleation

Response: We changed "*participating in*" to "*to*".

- L84, formation

Response: We changed "*formations*" to "*formation*".

- L137-138, I think 3080 include DMA inside

Response: We deleted "*, a differential mobility analyzer (DMA, model TSI 3081)*" in the sentence.

- L188&198, where, ???

Response: We deleted the comma after "*where*".

- L203, particles in new mode? What does this mean?

Response: We revised the whole paragraph as:

"*Days of NPF events was classified according to the method proposed by Dal Maso et al. (2005) and Kulmala et al. (2012), in which (a) a burst in the concentration of sub-3 nm particles or clusters was observed and (b) these particles had a continuous growth over a time span of hours (e.g., usually more than ten hours). If no clear growth of these newly formed particles (sub-3 nm particles) can be identified, the day was classified as an undefined day. The day without both the burst of sub-3 nm particles and their subsequent growth was considered as a non-event day.*"

- L207, discussion

Response: We changed it to "*Results and discussion*".

- L208, section 3.1.

Response: We added "*3.1*" before "*General characteristics of NPF at GC site*".

- L215, at the current site

Response: We added "*the*" before "*current site*".

- L222 & others, lower than, not "compared to"

Response: We changed "*compared to*" to "*than*".

- L226, what are they referred to?

Response: We changed "*They*" to "*Yue et al. (2009) and Wang et al. (2013)*".

- L253&others, "Note that" is simple than "It has to be noted that"

Response: We changed "*It has to be noted that*" to "*Note that*".

- L256-257, it is hard to understand this sentence

Response: We revised the sentence to:

"*It has to be noted that most atmospheric formation rate reported in China was based on the measured formation rate at relatively larger size, i.e., 3-10 nm, which is so called the "apparent" particle formation rate. In order to derive the formation rate of critical clusters from the "apparent" particle formation rate (Kulmala et al., 2017), the nuclei GR or GR at sub-3 nm is needed but usually remains unclear.*"

- L262, formation rates vs those

Response: We used "*formation rates*" in the revised manuscript.

- L266, in those polluted atmospheres

Response: We changed "*in those polluted atmosphere*" to "*in those polluted atmospheres*".

- L288-289, that clause is ambiguous here

Response: We revised the sentence to:

"*This could be attributed by the high CS or CoagS at those polluted environments as the growth of small particles is limited, which are more vulnerable to the coagulation scavenging.*"

Reference

Cai, R., Yan, C., Worsnop, D. R., Bianchi, F., Kerminen, V-M., Liu, Y., Wang, L., Zheng, J., Kulmala, M., & Jiang. J., (2021) An indicator for sulfuric acid–amine nucleation in atmospheric environments, Aerosol Science and Technology, 55:9, 1059-1069, DOI: 10.1080/02786826.2021.1922598.

Lu, Y., Yan, C., Fu, Y., Chen, Y., Liu, Y., Yang, G., et al. (2019). A proxy for atmospheric daytime gaseous sulfuric acid concentration in urban Beijing. Atmospheric Chemistry and Physics, 19(3), 1971–1983. https://doi.org/10.5194/acp-19-1971-2019.

---

## Author Response (AR2)

**Response to the editor:**

Comment from the editor: thanks for the detailes reply to the reviewers. I just read it and I am almost satisfied with the result. There is one point in the answer to reviewer 2. You show a figure, named Figure 4 and in the caption circles and squares are mentioned. But I am not able to say for most of the symbols if it is a circle or a square. In the modified manuscript the figure appears as part of figure 5, if I am not wrong and for this figure b) nothing about circles and squares is written. Can please clarify this?

Response: We would like to thank the editor for the detailed suggestion. We clarified this by adding more clear descriptions of these dots in Figure 5 for our manuscript. Meanwhile, we changed these squares to triangles and specifically for the right panel of Figure 5, we made their edge color black, trying to differentiate these triangles with the circles more obviously. The modified figure was shown below as well.

[Figure]

Figure 1: (a) $H_2SO_4$ concentration as a function of condensation sink during both event days and non-event days during our study. (b) The dimensionless indicator, $I$, as a function of the condensational sink. For both panels, the triangles indicate data for event days while the circles indicate data for non-event days. The colorbar indicates: solar radiation (left panel) and $H_2SO_4$ proxy concentration (right panel).